# Improved Reversion of Calcifications in Porcine Aortic Heart Valves Using Elastin-Targeted Nanoparticles

**DOI:** 10.3390/ijms242216471

**Published:** 2023-11-17

**Authors:** Anja Feldmann, Yvonne Nitschke, Franziska Linß, Dennis Mulac, Sina Stücker, Jessica Bertrand, Insa Buers, Klaus Langer, Frank Rutsch

**Affiliations:** 1Department of General Pediatrics, Muenster University Children’s Hospital, D-48149 Muenster, Germany; anjafeldmann22@gmx.de (A.F.); yvonne.nitschke@ukmuenster.de (Y.N.); insa.buers@ukmuenster.de (I.B.); 2International Network of Ectopic Calcification (INTEC), 9000 Ghent, Belgium; f_hove01@uni-muenster.de (F.L.); sina.stuecker@med.ovgu.de (S.S.); jessica.bertrand@med.ovgu.de (J.B.); 3Institute of Pharmaceutical Technology and Biopharmacy, University of Muenster, D-48149 Muenster, Germany; dennis.mulac@uni-muenster.de (D.M.); k.langer@uni-muenster.de (K.L.); 4Department of Orthopaedic Surgery, Otto-von-Guericke-University Magdeburg, D-39120 Magdeburg, Germany

**Keywords:** nanoparticles, aortic valve, calcification, targeted chelation therapy, anti-elastin antibody

## Abstract

Calcified aortic valve disease in its final stage leads to aortic valve stenosis, limiting cardiac function. To date, surgical intervention is the only option for treating calcific aortic valve stenosis. This study combined controlled drug delivery by nanoparticles (NPs) and active targeting by antibody conjugation. The chelating agent diethylenetriaminepentaacetic acid (DTPA) was covalently bound to human serum albumin (HSA)-based NP, and the NP surface was modified using conjugating antibodies (anti-elastin or isotype IgG control). Calcification was induced ex vivo in porcine aortic valves by preincubation in an osteogenic medium containing 2.5 mM sodium phosphate for five days. Valve calcifications mainly consisted of basic calcium phosphate crystals. Calcifications were effectively resolved by adding 1–5 mg DTPA/mL medium. Incubation with pure DTPA, however, was associated with a loss of cellular viability. Reversal of calcifications was also achieved with DTPA-coupled anti-elastin-targeted NPs containing 1 mg DTPA equivalent. The addition of these NPs to the conditioned media resulted in significant regression of the valve calcifications compared to that in the IgG-NP control without affecting cellular viability. These results represent a step further toward the development of targeted nanoparticular formulations to dissolve aortic valve calcifications.

## 1. Introduction

Calcific aortic valve disease (CAVD) with its end stage of calcific aortic valve stenosis is one of the most common heart conditions in industrialized countries. While the prevalence of this condition is high in the elderly, there are a few rare hereditary syndromes such as Singleton-Merten syndrome, Hutchinson-Gilford-Progeria, or Gaucher-disease type IIIC, which are accompanied by calcification of the aortic valve in early childhood itself [1,2,3,4].

The aortic valve as one of the four valves of the heart guarantees a unidirectional blood flow from the left ventricle towards the aorta [5]. The valve itself needs to resist high shear forces, and therefore, a closer look needs to be taken at the microstructure of the aortic valve [6]. Each of the three leaflets of the aortic valve consists of three histological layers. The fibrosa faces towards the aorta, while the ventricularis faces towards the left ventricle [7]. In between these two layers, there is a gelatinous layer called spongiosa. Additionally, the valve is covered by valve endothelial cells (VECs), which form a protection barrier around the valve [7]. The main components of the aortic valve extracellular matrix (ECM) are collagen, elastin, glycosaminoglycans, and proteoglycans, which mainly get secreted by valve interstitial cells (VICs) existing in all valve layers [8,9,10,11].

Elastic fibers, as part of the valve’s ECM, consist of an amorphous elastin core that is cross-linked with microfibrils (i.e., fibrillin or fibulin-4/5), glycosaminoglycans, proteoglycans, and the elastin receptor. Elastic fibers are located in tissues that need to resist high shear forces and are also found in the skin, arteries, lungs, gall bladder, Bruch’s membrane, and vocal cords to guarantee elasticity [8,11,12]. While elastin is a durable protein that lasts an entire lifetime, damage of elastin, e.g., due to metabolic factors, leads to the activation of elastases and matrix metalloproteinases and therefore the release of elastokines [13]. These elastokines exhibit high reactivity and often initiate inflammation and calcification. Elastic fibers can bind calcium due to calcium-binding glycoproteins, charged carboxyl- and amino groups, and neutral binding sites [12,14,15]. Therefore, due to the release of elastokines after degradation, elastin often is associated with the origin of pathologic calcifications [12].

The calcification of the aortic valve underlies a complex pathophysiology. It is an actively regulated process in which calcium and phosphate are deposited within the aortic valve and restrict the function of the heart valve [14]. Many factors can lead to CAVD, including metabolic, degenerative, inflammatory, or genetic factors. Furthermore, an imbalance of pro or non-calcifying factors can lead to the development of CAVD [3,4,16].

CAVD is characterized by two phases: the initiation phase and the propagation phase [4]. The initiation phase starts with primary endothelial damage, which forms an entry port for low density lipoproteins (LDLs), lipoprotein(a) (Lp(a)), and monocytes. This initiates chronic inflammation that leads to the activation of VICs [4,17]. The activation of VICs initiates the second phase; VICs release calcifying vesicles, inflammation becomes stronger, and calcifications aggregate within the valve’s tissue. In CAVD, calcification is mainly found in the outer histological layers due to endothelial lesions. However, the ventricularis, with its high content of elastin and the capability to bind calcium, presents with strong calcifications [4,12,17,18].

The more the valve calcifies, the smaller the opening area of the valve becomes. This end stage of the CAVD, aortic valve stenosis, has a high prevalence in the elderly and causes major health problems. The only treatment option nowadays is the replacement of the valve with a new one as no drug therapy exists [19,20]. Surgical replacement of the aortic valve is a frequent procedure; however, surgery poses significant risks for the patient [19]. The aortic valve either gets replaced via open heart surgery or transcatheter aortic valve implantation (TAVI), both potentially associated with various perioperative complications.

Recently, the use of chelating agents to treat cardiovascular diseases has been brought into focus. Chelating agents such as ethylenediamine tetraacetic acid (EDTA), dimercaprol, dimercaptosuccinic acid (DMSA), dimercapto-propane sulfonate (DMPS), and diethylenetriamine pentaacetic acid (DTPA) are capable of building up coordinative bindings with cations [21,22,23]. While chelation therapy is the gold standard in the treatment of heavy metal poisoning, chelators themselves bear the risk of severe side effects such as hypocalcemia, renal dysfunction, or neurological disorders [24]. The “Trial to Assess Chelation Therapy” (TACT) from 2003–2012 revealed modestly reduced cardiovascular events in stable diabetic or post-myocardial infarction patients after intravenous treatment with EDTA, vitamins, and minerals. The TACT2 is still ongoing to confirm the results of the TACT [25,26]. The TACT remains highly controversial, as its results are not fully evidence-based and are often criticized for the lack of information about the trial [26]. As targetless chelation therapy is known to go along with high complications, the lack of evidence in the results needs to be emphasized.

In various fields of medicine, including digital imaging or cancer therapy, nanoparticles (NPs) have been designed that specifically target cells or tissue and have become highly relevant for medicine. While the problem of chelation therapy is the lack of targeting specific tissue, chelation therapy and nanomedicine might get combined. Research has been performed on conjugating a chelator to a NP and further conjugating the NPs with an antibody, so that the chelator could act locally at a specific target [21,22,23].

In this study, we describe an effective way to use human serum albumin (HSA)-based NPs with the conjugated chelator DTPA for the reversion of ex vivo induced valvular calcification in porcine aortic valves. The NPs were able to target calcified sites within the valve correctly due to the conjugation of an anti-elastin antibody, while the viability of the valvular cells was preserved.

## 2. Results

### 2.1. Preparation and Characterization of Nanoparticles

The resulting characteristics of the antibody-coupled NPs and the extent of covalently bound DTPA are shown in Table 1. The hydrodynamic diameter of the NPs was in the size range of 160–170 nm. The uniformity of the size distribution is expressed by the polydispersity index, which with a value below 0.1 indicates a narrow size distribution. The zeta potential of the different formulations was above −30 mV, which is an indication of sufficient electrostatic stabilization of the colloidal dispersion. Indirect quantification by HPLC analysis determined the covalent binding of 9 mol DTPA per mol HSA.

### 2.2. Characterization of Ex Vivo Calcification of Porcine Aortic Valve Leaflets

Porcine aortic valve leaflets were cultured in an osteogenic medium for up to 10 days to induce ex vivo calcification. As the control, leaflets were cultured in the standard medium. In leaflets incubated in the standard medium, calcium content remained steady from day one till day 10 at 1.50–2.07 µg Ca^2+^/mg dry weight (Figure 1A), and histologically no calcification could be found (Figure 1B, upper two panels). Leaflets remained viable under standard culture conditions for the whole incubation period, comparable to native leaflets without culturing (Figure 1B, lower panel). Leaflets incubated in the osteogenic medium presented an increase in calcium content from 1.97 ± 0.73 µg Ca^2+^/mg dry weight on day one to 5.29 ± 1.9 µg Ca^2+^/mg dry weight on day 10 (*p* ≤ 0.01; Figure 1A). Histologically, calcification infiltrated from the outer layers towards the inner layers and became denser with a longer incubation period (Figure 1B, upper two panels). Compared to aortic valve leaflets incubated in the standard medium, viability was reduced after five days in the osteogenic medium. On day 10, the viability further decreased to approximately half (Figure 1B, lower panel). To identify the type of calcium crystals that deposit in calcified aortic valves cultured in the osteogenic medium for 10 days, we employed confocal Raman spectroscopy. This method allows precise identification of different types of calcium crystals based on the characteristic peaks in the Raman shift, e.g., at 960 cm^−1^ for basic calcium phosphate (BCP) or at 1050 cm^−1^ for calcium pyrophosphate (CPP) crystals. Von Kossa-stained sections were used to visualize calcified aortic valve tissue and guide ROI selection for Raman mapping (Figure 2A). Scanning these calcified areas, we exclusively detected BCP crystal deposition (Figure 2B). As control, a measurement of a comparable location with similar cell density of a non-calcified aortic valve, cultured for 10 days in the standard medium, was performed. Here, only background noise without any RAMAN signal at 960 cm^−1^ was observed (Appendix A).

### 2.3. Ex Vivo Targeting of Elastin Antibody-Labeled NPs within Aortic Valve Leaflets

Aortic valve leaflets were pre-cultured for five days under standard or osteogenic conditions. PromoFluor 633P-labeled NPs conjugated with anti-elastin antibody (HSA-NP-(PF633)-AntiElastinPig) or as control PromoFluor 633P-labeled NPs conjugated with IgG (HSA-NP-(PF633)-IgG) were incubated for four hours with the aortic valve leaflets (50 µg NP/mL medium). HSA-NP-(PF633)-AntiElastinPig were bound to the aortic valve leaflets targeting elastin-rich areas and were able to infiltrate the valve, whereas control HSA-NP-(PF633)-IgG did not bind to the aortic valve leaflets (Figure 3A,B). HSA-NP-(PF633)-AntiElastinPig did not bind to uncalcified aortic valves, cultured in the standard medium (Figure 3A).

### 2.4. DTPA-Loaded NPs Reversed Induced Calcification in Aortic Valve Leaflets Ex Vivo

To reverse existing calcification, aortic valve leaflets were pre-incubated for five days in an osteogenic medium. In pretests, we found that a concentration of 1 mg/mL pure DTPA incubated for two days resulted in a reduction of calcium content of pre-calcified valves (0.72 ± 0.11 µg Ca^2+^/mg dry weight versus osteogenic medium control 5.16 ± 1.25 µg Ca^2+^/mg dry weight; *p* ≤ 0.01), while addition of 0.5 mg/mL DTPA did not lead to any reversion of calcium content (5.15 ± 0.46 µg Ca^2+^/mg dry weight. Based on these results, on day five, the pre-incubated aortic valve leaflets were treated with equivalent NPs (DTPA-(HSA-NP)-AntiElastinPig), which contained the same DTPA concentration as 1 mg/mL pure DTPA, for 2 days. As control, leaflets were incubated with 1 mg/mL pure DTPA, DTPA-(HSA-NP)-IgG as the isotype control, or solely osteogenic medium. It was shown that DTPA-(HSA-NP)-AntiElastinPig was able to significantly reverse calcifications (0.90 µg ± 0.13 Ca^2+^/mg dry weight, versus osteogenic medium control 5.16 µg ± 1.25 Ca^2+^/mg dry weight, *p* ≤ 0.001, Figure 4A,B (upper two panels). Further, 1 mg/mL pure DTPA and DTPA-(HSA-NP)-IgG were successful in reversing existing calcifications as well (0.72 µg ± 0.11 Ca^2+^/mg dry weight (*p* ≤ 0.001) and 1.99 µg ± 0.15 Ca^2+^/mg dry weight (*p* ≤ 0.001) versus osteogenic medium control 5.16 µg ± 1.25 Ca^2+^/mg dry weight, respectively; Figure 4A,B (upper two panels). DTPA(HSA-NP)-AntiElastinPig and pure DTPA had a similar effect on dissolving calcification (no significant difference). However, with DTPA-(HSA-NP)-IgG, a lower decrease in the calcium content was achieved compared to DTPA-(HSA-NP)-AntiElastinPig and pure DTPA (*p* ≤ 0.05, both). Pure DTPA was highly effective in reversing calcifications; however, pure DTPA and DTPA-loaded NPs presented different results in the viability. The addition of pure DTPA resulted in a major decrease in cellular viability, whereas the two NP formulations did not affect it (Figure 4B, lower panel).

## 3. Discussion

CAVD is associated with fibrosis, thickening, and mineralization of aortic valve leaflets. Ectopic mineralization co-occurs with osteogenic transition of VIC, microhemorrhage, and deposition of abnormal extracellular matrix. Elevated plasma lipoprotein(a) levels, diabetes mellitus, and obesity have been identified as risk factors, and disordered signaling, including NOTCH, myocardin, and WNT-beta-catenin pathways, seem to be strongly involved in the progression of fibrosis and calcification (for review, refer to [27]). In CAVD, calcification of the aortic valve, which usually guarantees a unidirectional systemic blood flow, limits the proper functioning of the heart valve connecting the left ventricle with the aorta [10,16,20]. The progress of calcification can cause major health problems from dyspnea to syncope. In the CAVD end stage, the so-called aortic valve stenosis, surgical intervention remains the only treatment option [19,28]. Until today, there is no drug therapy for CAVD/aortic valve stenosis, and concomitant medication currently only treats comorbidities or attempts to improve symptoms. While surgical intervention accompanied by its risks is the established cure for CAVD, new investigations in the field of nanomedicine and targeted drug delivery bring along promising findings for drug therapy options in reversing calcification in CAVD [21,23]. Targeted drug delivery in the field of nanomedicine, as it is already present in cancer therapy or for imaging techniques, follows the idea of directing a drug to a particular target, e.g., via conjugation of a specific antibody. Combining these new ideas of establishing a targeted drug delivery-based medication for cardiovascular diseases, chelating agents including EDTA or DTPA bring along the advantage of chelating cations [21,23,29,30]. This might have a positive impact on cardiovascular diseases.

In our study, we established a tissue culture set-up using porcine aortic valve leaflets that were stimulated to calcify ex vivo. The use of ex vivo valve leaflet tissue culture has the advantage of the VICs being preserved in the physiological arrangement regarding their connection with the ECM and their relationship with the VEC.

Although the tissue structure of porcine aortic valves is highly comparable to human aortic valves, there are still differences in the microstructure of the valves, which limit the transferability to human aortic valves [31]. However, porcine aortic valves from a commercial abattoir, which will be wasted anyway, are a good choice for basic studies and screening of inhibitors. In this porcine aortic valve ex vivo model, we were able to reverse existing calcifications using DTPA-loaded HSA-based NPs with a conjugated anti-elastin antibody. The antibody modification of the NPs allowed specific targeting of elastin in the aortic valve tissue.

In a previous study of our group, the effect of these NPs has already been successfully demonstrated in an ex vivo model on murine aortic ring sections. The similarly prepared coupled-NPs showed significant inhibition or reversion of tissue calcification [21]. In the current study, the obtained physicochemical characteristics of the NPs with regard to hydrodynamic diameter (160–170 nm) and polydispersity index (<0.1) were comparable. The covalently bound amount of DTPA was also within the comparable range at about 9 mol DTPA per mol HSA. Moreover, the zeta potential, which was above −30 mV in all tested NP formulations, is a strong indicator of sufficient NP stabilization by electrostatic repulsion [32]. Therefore, it is possible to draw conclusions about the suitability of these NPs in terms of stability and cytotoxicity, which have been adequately examined previously [21].

Induction of calcification of porcine aortic valve leaflets required an elevated inorganic phosphate (P_i_) concentration (2.5 mM P_i_). Valve leaflets cultured under osteogenic conditions maintained viability, although reduced, for at least 10 days. Using Raman spectroscopy, mainly basic calcium phosphate (BCP) crystals were identified in the calcified areas of aortic valves, whereas non-calcified valves displayed any Raman signal at 960 cm^−1^. The relatively low Raman signal intensity found in the calcified valves may be due to the diffuse and weak calcification already observed in the von Kossa staining. In addition, calcified deposits may be embedded within the tissue section, which could impair signal intensity. Our measurements were two-dimensional, only scanning the surface of the tissue section, so deeper calcified deposits may show a weak signal. Three-dimensional measurements penetrating the full tissue depth might improve signal intensity. Calcification infiltrated from the outer valvular layers towards the inner. While in the human body, the fibrosa is more prone to mineralization compared to the ventricularis, in our study, the ventricularis was often the first layer to calcify [3,33]. It is suggested that the high shear forces due to systemic blood circulation might ablate calcifications on the ventricularis, as this is the layer that faces toward the left ventricle [33]. In our static ex vivo aortic valve culture, no shear forces occurred. The aortic valve functions in a complex mechanical environment that includes stretching, bending, pressure, and shear forces [34]. Changes in these factors have been associated with the development and progression of valve malfunction and CAVD [35,36]. However, it was shown that even inducing shear stress via an accelerated pulse tester in bovine and porcine heart valve tissue led to a pattern of calcification that was individual from valve to valve [37]. A more complicated ex vivo porcine aortic valve tissue culture requires incubation in a stretch bioreactor for up to two weeks in an osteogenic medium supplemented with high phosphate. With this model, authors could achieve valve calcification in response to a pathologically relevant level of cyclic stretch and osteogenic medium within a week, preferentially on the fibrosa side [38]. Using our model under static conditions without stretching, valves might enroll. We did not observe an increase in convolving of the valve leaflets under different conditions. However, enrolment may lead to an insufficient medium supply, leading to a decrease in degeneration and a loss of layer-dependent progression of calcification [39]. Enrolment could be inhibited by pinning the valves to the base of the tissue culture plate [39,40]. But even pinning did not always result in the specific calcification pattern within the valve seen in CAVD, i.e., the calcification was not primarily associated with the fibrosa of the valve leaflets [40]. Authors further observed a tendency of increased calcification at the ends of the valves, which might be due to the pinning of the valve to the culture plate [40]. The porcine aortic valve cultured ex vivo only in an osteogenic medium will never develop like the human calcified aortic valves in CAVD. In CAVD, changes developed over several years including inflammation and fibrosis, whereas calcification in our model developed in only 5 days. However, our study focused only on the development of a simple ex vivo aortic valve calcification model to investigate the resorption of existing calcifications using NPs in tissue.

The viability of the porcine aortic valve leaflet cells of our study was found to be reduced in an osteogenic medium compared to the standard medium. While the cells remained fully viable in the standard medium, viability decreased over time during incubation in the osteogenic medium to approximately 50% after 10 days. Interestingly, in other studies, it was shown that incubation of porcine aortic valves in a different osteogenic medium containing more P_i_ (3.8 mM) did not result in any changes in viability over 8 days [41].

We showed success in resolving BCP crystals in aortic valves ex vivo via DTPA in a concentration of 1 mg/mL. Comparing the success of DTPA, EDTA, and sodium thiosulfate (STS) in chelating calcium, EDTA and DTPA were found to be the most effective; DTPA was able to remove calcium at a concentration of 1 mg/mL DTPA [22]. Smaller amounts of DTPA were used to remove calcium from calcified murine aortic rings ex vivo in a previous study; 0.014 mg/mL was already sufficient to reverse calcifications [21]. The difference in the tissue size and surface area of a murine aorta and a porcine aortic valve might be an explanation for the different concentrations needed.

The idea of using chelating agents in the treatment of cardiovascular diseases poses possibilities and risks at the same time. The TACT demonstrated a modestly reduced risk for cardiovascular events in stable diabetic/post-myocardial infarction patients after intravenous treatment with EDTA, vitamins, and minerals. The TACT2 is an ongoing trial in the hope of supporting the hypothesis raised in the TACT [25,26]. This is a highly controversial trial that has been criticized from many sides as the study design of the TACT had several limitations [42]. Chelators can chelate divalent cations, and when administered systemically, cations can be chelated aimlessly. In the context of systemic chelation therapy, hypocalcemia, bone loss, or death count are the consequences reported [42,43,44]. Accordingly, in our ex vivo model, we could observe that administration of 1 mg/mL pure DTPA resulted in a complete loss of cellular viability.

At this point, targeted chelation therapy has become highly relevant. The combination of conjugating NPs with antibodies and a chelator at the same time forms a safer way to avoid side effects associated with the systemic administration of chelators. In our study, we showed that NPs conjugated with anti-elastin antibodies were able to target the elastic fibers of porcine aortic valves, while the embedded drug (DTPA) could significantly chelate calcium alongside the calcified elastic fibers. To our knowledge, we are the first to show a positive effect on the reversion of ex vivo calcification within aortic valves using NP as most other studies have investigated the reversion of arterial calcification [23,30,45]. We found that the anti-elastin antibody-bound NPs in concentrations of 50 µg/mL to the target only already calcified tissue, while uncalcified healthy tissue was not targeted. This specific binding was also achieved by others in vivo using specific antibodies against core elastin, which only gets revealed once elastic fibers get damaged [23,46,47]. This specific antibody ensures that healthy elastin will not be targeted by the NPs. The antibody used in this study was not exclusively directed to core elastin. However, elastin is woven into the valvular tissue, and access to the elastic fibers is only possible if the elastic fibers are damaged, for example, by calcification. In healthy aortic valves, the surface of elastin fibers is coated with microfibrils mainly composed of fibrillin, which binds microfibrillar glycoproteins, fibulins, and microfibrillar-associated proteins; therefore, elastic fibers are most likely not accessible to elastin-targeted antibodies in healthy tissues [48,49,50]. Only during calcification, elastic fibers become fragmented and damaged due to enzymatic degradation of microfibril proteins, exposing them to antibodies directed against elastin. DTPA-(HSA-NP)-IgG used as control NPs, did not connect to the valvular tissue, while they were still effective in chelating calcium without connecting to the valve. The chelating ability of the DTPA-(HSA-NP)-IgG was less effective than the chelating ability of the DTPA-(HSA-NP)-AntiElastinPig, as the NPs with the specific antibody directed against elastin could connect to the valve’s elastic fibers, the location where calcifications were present. As a result, DTPA came into close contact to the valve tissue and is released from the NP matrix over time, allowing subsequent tissue penetration. We consider that the effect of DTPA-(HSA-NP)-IgG on the resolution of calcification was mainly due to the release of DTPA from the NP into the tissue culture medium instead of non-specific binding. In addition to the ability to dissolve existing calcifications by DTPA and DTPA-loaded NPs, it must be emphasized that the use of DTPA-loaded NPs, in contrast to pure DTPA, offers a major advantage; while 1 mg/mL of pure DTPA resulted in a total loss of aortic valve viability, the same amount of DTPA conjugated to NPs guaranteed full cellular viability. Additionally, we observed that the addition of 1 mg/mL pure DTPA further decreased the pH of the culture medium from 7.3 in the osteogenic medium to 7.1. This decrease in pH might have a further impact on cell death in pure DTPA-treated valves. Interestingly, pH was not altered after the addition of NP formulations. Taken together, this highlights the strength of NPs in their potential use as a safe therapeutic option for ectopic soft tissue calcification with targeted drug delivery via NPs. While our NPs were based on human serum albumin, a highly biodegradable and non-toxic matrix, with the conjugation of an antibody and the chelator DTPA, there are various other ways to design NPs that can interfere with soft tissue calcifications. Nosoudi et al. developed human serum albumin-based NPs targeted to damaged elastin that were conjugated with pentagalloyl glucose (PGG), a plant-derived polyphenolic tannin, that forms a protective layer around elastin once connected to the elastic fibers [30,51]. They showed that these NPs were effective in resolving calcifications in the aorta in rats in vivo. We believe that this has an important role in the resolution of cardiovascular calcification, which can be achieved with chelators containing NPs as shown by us and others [21,23,30,45]. Additionally, some studies using NPs containing different substances showed effective inhibition or prevention of cardiovascular calcifications in vitro or in vivo [52,53].

Although tissue culture experiments do not recapitulate in vivo conditions, the study of ex vivo calcification of porcine aortic valves may yield insights into pathologic processes. Additionally, it is a simple method for screening potential therapeutic compounds. We are the first to have proven that ex vivo induced valvular calcification in porcine aortic valves could be reversed via targeted chelation therapy. An HSA-based NP system with a conjugated chelator DTPA and a conjugated anti-elastin antibody were used and showed promising results in treating valvular calcification ex vivo while preserving the viability of the aortic valve cells. This study might pave the way to further investigate nanoparticle compounds for use in CAVD in the future.

## 4. Materials and Methods

### 4.1. Nanoparticle Preparation

The nanoparticle formulations used in this study were prepared by established methods, which were also used and precisely described in a previous publication by our group [21]. In the following, the manufacturing process of the HSA-based nanoparticles (HSA-NP) is presented in brief.

The core HSA-NP were prepared using a desolvation method [54]. Therefore, 1 mL aqueous HSA solution (50 mg/mL) was adjusted to pH 7.5 with 2 N NaOH. A quantity of 4 mL ethanol 96% (*v*/*v*) was added dropwise to achieve the NP formation. To stabilize the HSA-NP by cross-linking, 11.6 µL glutaraldehyde 8% (*m*/*v*) was added followed by stirring overnight (550 rpm, 22 °C). The resulting HSA-NP was purified twice via centrifugation (16,000× *g*, 15 min) and resuspension in a medium suitable for subsequent modification.

For fluorescent labelling, promofluor 633P (PF633, PromoCell GmbH, Heidelberg, Germany) labelled HSA was prepared according to a method previously described [55]. The NPs were prepared as described above, with a proportion of 20% of the HSA replaced by labelled HSA-PF633.

To achieve covalent binding of DTPA to HSA-NP, a method described by Keuth et al. [21] was used. Therefore, 9.08 mg DTPA dianhydride (Thermo Fisher, Karlsruhe, Germany) was added to 41.3 mg HSA-NP dispersed in NaHCO_3_ buffer 0.05 M at pH 7.0. After 1 h of stirring (550 rpm, 22 °C), unbound DTPA was removed via 2-fold centrifugation (16,000× *g*, 15 min), and the resulting DTPA-(HSA-NP) was redispersed in a phosphate buffer at pH 8.0 for further antibody conjugation.

To functionalize the NPs by antibody binding, reactive thiol groups were introduced to the NP surface. For this purpose, 2-iminothiolane was added to the NP suspension in 50-fold molar excess. Incubation was carried out in a phosphate buffer at pH 8.0 (24 h; 22 °C; 600 rpm) followed by two purification steps (centrifugation; 16,000× *g*; 15 min) and redispersion in a phosphate buffer at pH 8.0.

For antibody modification of the NPs, in the first step, the respective antibody anti-elastin (mouse monoclonal ab9519, abcam, Cambridge, UK) or IgG (rabbit serum, Sigma Alrich, Steinheim, Germany) was coupled with the heterobifunctional crosslinker NHS-PEG_5000_-Mal (Rapp Polymere, Tübingen, Germany). The crosslinker was added in 15-fold molar excess to the antibody amount, and the mixture was incubated for 1 h (22 °C; 600 rpm). The resulting thiolated NPs and the PEG antibody conjugate were combined in the final step and incubated overnight (22 °C; 600 rpm). Subsequently, the final NPs were purified once by centrifugation (12,000× *g*; 12 min) and redispersion in purified water.

### 4.2. Nanoparticle Characterization

Hydrodynamic particle diameter and polydispersity index (PDI) were determined by photon correlation spectroscopy (PCS). The measurement of the diluted aqueous NP suspension was performed at 22 °C using a backscattering angle of 173°. The same NP dilution was used for surface charge characterization by zeta potential measurement. All measurements were conducted with a Malvern Zetasizer Nano ZS system (Malvern Instruments Ltd., Malvern, UK).

The indirect quantification of the NP-bound DTPA content was carried out by HPLC analysis as previously described by Keuth et al. [21]. The detection of DTPA was carried out after complexation with iron (III)-chloride. The complex was detected at 257 nm with an HPLC-DAD system (Agilent Technologies 1200 series, Agilent, Santa Clara, CA, USA).

### 4.3. Porcine Heart Preparation and Tissue Culture

Porcine hearts were provided by a local abattoir. After refrigerated transport, the porcine aortic valves with its three leaflets were obtained. An average leaflet weighs about 32 mg and measures 2 cm × 1.3 cm. The aortic valve leaflets were randomized between experimental groups. Aortic valve leaflet culture was modified from published ex vivo culture models [41,56]. Leaflets were transferred into 1 mL standard medium (DMEM medium (GE healthcare, Chalfont St. Giles, UK) supplemented with 10% FCS gold (Biochrom, Berlin, Germany) and 100 U/mL penicillin and 100 µg/mL streptomycin) in 12 well plates for static incubation at a temperature of 37 °C with 5% CO_2_. Whole leaflets were cultured in separate wells. The day after heart preparation was defined as day 0 of further experiments. The medium was changed every other day, and no evaporation of the medium had taken place between the change intervals. The leaflets were harvested on the day of interest and rinsed once in PBS. For histological assessment, leaflets were cut in half approximately in the middle of the tissue and then cryo-embedded, cut side down. For calcium quantification, leaflets were dried for 48 h at 60 °C. Calcification was induced by incubation in an osteogenic medium (standard medium supplemented with 2.5 mM P_i_). Addition of 2.5 mM P_i_ decreased the pH from 7.5 in the standard medium to pH 7.3 in the osteogenic medium after 10 min at 37 °C and 5% CO_2_.

For reversion experiments, leaflets were pre-incubated in the osteogenic medium starting from day 0 to induce calcification. On day five, the pre-warmed medium was supplemented with DTPA at a concentration of 0.05 or 1 mg/mL or the same DTPA amount bound to NPs (DTPA-(HSA-NP)-AntiElastinPig or DTPA-(HSA-NP)-IgG) to reverse existing calcifications. Addition of 1 mg/mL DTPA to the osteogenic medium decreased the pH from 7.3 to 7.1 after 10 min at 37 °C and 5% CO_2_. Addition of NP formulations did not change the pH in the osteogenic medium.

### 4.4. Viability of the Porcine Aortic Valve Leaflet Cells

Viability was determined by the incubation of leaflets in the culture medium supplemented with 0.5 mg/mL MTT (methylthiazolyldiphenyl-tetrazolium bromide) for three hours. After rinsing the leaflets once with PBS, leaflets were embedded in the same way as for histological assessment. Pre-cutting was performed starting at the middle cut side of the leaflet, which was facing downwards during embedding, until the tissue was completely visible on the slide. From this moment, the collection of the slices started (serial sections of 14–16 µm). Every tenth section was analyzed approximately in the middle of the leaflet, three sections per leaflet. Sections were investigated for blue staining, indicating viable cells.

### 4.5. Calcium Quantification

For calcium quantification, whole leaflets were dried for 48 h at 60 °C. Dried leaflets were incubated in 0.6 N HCl overnight to solubilize phosphocalcific deposits. The calcium content of the supernatants was determined by a previously described o-cresolphthalein complexone method [57]. Calcium content was determined at a wavelength of 570 nm and normalized on dry weight.

### 4.6. Histology and Immunofluorescence Staining Methods

Stainings were performed on cryo-embedded porcine aortic valve leaflets. Pre-cutting and collection of slides were done according to the leaflets used for viability staining. For visualization of calcified tissue, von Kossa staining was performed as previously described [58]. Every tenth section was analyzed approximately in the middle of the leaflet, similar to the viability assessment, three sections per leaflet.

For immunofluorescence staining of elastic fibers, slides were washed in tris-buffered saline (TBS; 25 mM Tris, 140 mM NaCl, 2.7 mM KCl, pH 7.4) and fixed for 15 min in ice-cold methanol afterward. The slides were pre-treated with 2 mg/mL hyaluronidase in 0.5 M sodium acetate buffer at pH 6.0 in a humidity chamber at 37 °C for 30 min. After washing, slides were blocked with 2% BSA/TBS for 1 h. Primary mouse monoclonal anti-elastin antibody (1:1000; ab9519, abcam, Cambridge, UK) was incubated at 4 °C overnight. Slides were then washed once with TBS, and the secondary antibody (donkey anti-mouse IgG, cy3, 1:50; Chemicon, Darmstadt, Germany) was added for 1 h; after another rinse with TBS and distilled water, the slides were covered with fluorescent mounting medium.

### 4.7. Raman Spectroscopy of Aortic Valves

Calcium crystals in aortic valves were identified by confocal Raman microscopy (Bruker Senterra II under OPUS 7.8, Bruker, Karlsruhe, Germany), as previously described [59]. Prior to mapping, cryo-cut sections were thawed, fixed with methanol, and air dried. Overview images of dried samples were taken with a 10× objective to select ROI for mapping. Complementary von Kossa images were used to guide ROI selection. Raman spectra were obtained on a grid of 3 µm^2^ using a 785 nm laser with 50 mW power and a 20× objective. Measurements were performed with an integration time of 500 ms and a spectral resolution of 1.5 cm^−1^, covering a spectral range of 50–1410 cm^−1^ to detect calcium crystal-specific peaks at 960 cm^−1^ for BCP and 1050 cm^−1^ for CPP, respectively. Outliers such as cosmic rays were excluded manually. For data analysis, acquired spectra were normalized and integrated at 945–975 cm^−1^ and 1035–1060 cm^−1^ to detect BCP and CPP crystals, respectively, subsequently creating corresponding heat maps displaying calcium crystal distribution.

### 4.8. Statistics

All experiments requiring statistics were performed in biological triplicate and technical duplicate. Obtained results are presented as average values with standard deviation. For statistical analyses, one-way ANOVA on separated datasets was used, followed by Tukey’s post hoc test for comparisons between multiple groups via GraphPad Prism Version 9.1.1. Significance levels were declared as * for *p* ≤ 0.05, ** for *p* ≤ 0.01, and *** for *p* ≤ 0.001.

## Figures and Tables

**Figure 1 ijms-24-16471-f001:**
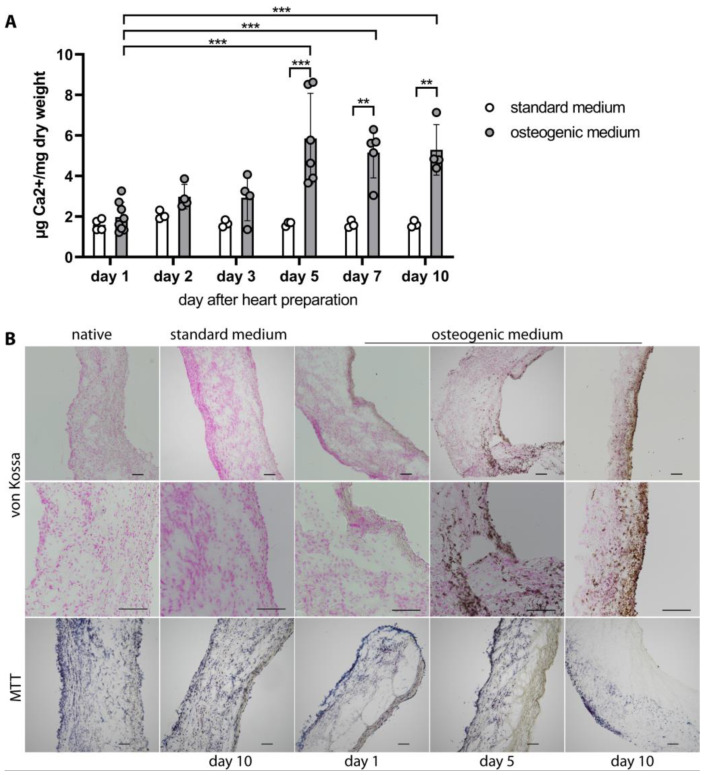
Induced calcification of porcine aortic valve leaflets. Leaflets were incubated in standard medium or osteogenic medium for indicated days. Leaflets incubated in standard medium did not calcify, whereas leaflets incubated in osteogenic medium presented an increase in calcification. (**A**). Calcium quantification, mean ± SD; *n* ≥ 3; ((**B**), upper two panels). Representative images of von Kossa staining of aortic valve leaflets visualizing calcified areas (brown). The ventricularis is presented on the right side of the images. As control, a native valve leaflet without culturing is shown (the middle panel represents a magnification of the upper panel, scale bars represent 100 µm; *n* ≥ 3 valve leaflets, *n* ≥ 3 sections per leaflet analyzed). ((**B**), lower panel). Leaflets were viable under standard culture conditions for the whole incubation period, while viability in valves cultured in osteogenic medium decreased. As control, normal viability of a native valve leaflet without culturing is shown (MTT (Methylthiazolyldiphenyl-tetrazolium bromide) staining, scale bars represent 100 µm; *n* ≥ 3 valve leaflets, *n* ≥ 3 sections per leaflet analyzed). ** *p* ≤ 0.01; *** *p* ≤ 0.001.

**Figure 2 ijms-24-16471-f002:**
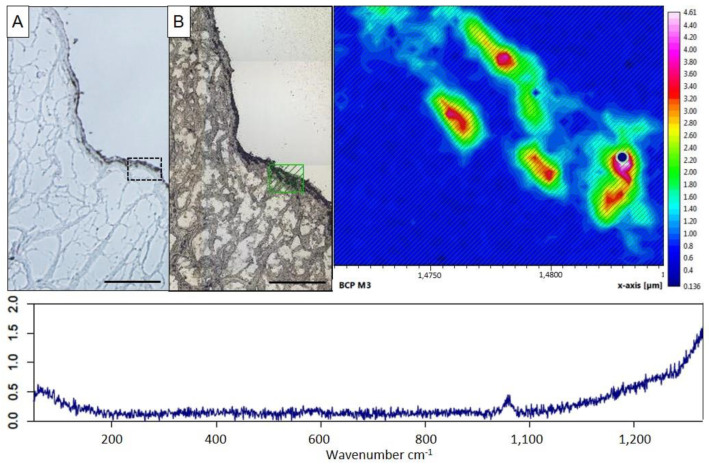
Representative Raman mapping of calcified aortic valves cultured in osteogenic medium for 10 days. (**A**). Complementary von Kossa image of this section displaying calcified deposits with selected ROI (black square) (upper left panel, scale bar: 200 µm). (**B**). Representative overview image of this section as unstained, methanol-fixed, and air-dried aortic valve with selected ROI (green square) (upper middle panel). Representative Raman spectrum of the indicated ROI displaying the characteristic peak for BCP at 960 cm^−1^ (lower panel, *Y*-axis shows the intensity of the scattered light (arbitrary units)). Representative heat map of BCP calcification after normalization and integration at 960 cm^−1^, showing the distribution of BCP crystals in the tissue (as determined by the integrated area under the curve at 945–975 cm^−1^, right panel). *n* = 2 valve leaflets, 4 sections per valve, 3–5 analyzed locations each.

**Figure 3 ijms-24-16471-f003:**
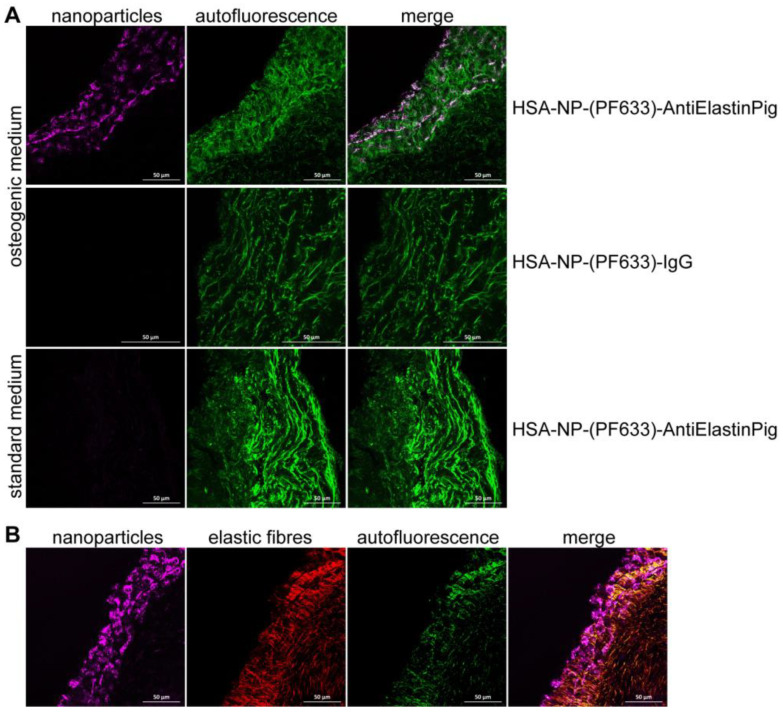
Visualization of targeting of PromoFluor 633P-labeled NPs at porcine aortic valve leaflets. Aortic valve leaflets were pre-cultured for five days under standard or osteogenic conditions prior to four hours of incubation with NP. (**A**). NP conjugated with anti-elastin antibody (HSA-NP-(PF633)-AntiElastinPig) connected to the calcified aortic valve leaflets (osteogenic medium) and were able to infiltrate the valve, while noncalcified valves (standard medium) were not targeted. Control NPs conjugated with IgG (HSA-NP-(PF633)-IgG) could not be detected at the calcified aortic valve leaflets (*n* ≥ 3 valve leaflets, *n* ≥ 3 sections per leaflet analyzed). (**B**). NPs conjugated with anti-elastin antibody (HSA-NP-(PF633)-AntiElastinPig) targeted to elastin-rich areas in the aortic valve. Nanoparticles: pink, autofluorescence: green, elastin: red. Scale bar: 50 µm (*n* ≥ 3 valve leaflets, *n* ≥ 3 sections per leaflet analyzed).

**Figure 4 ijms-24-16471-f004:**
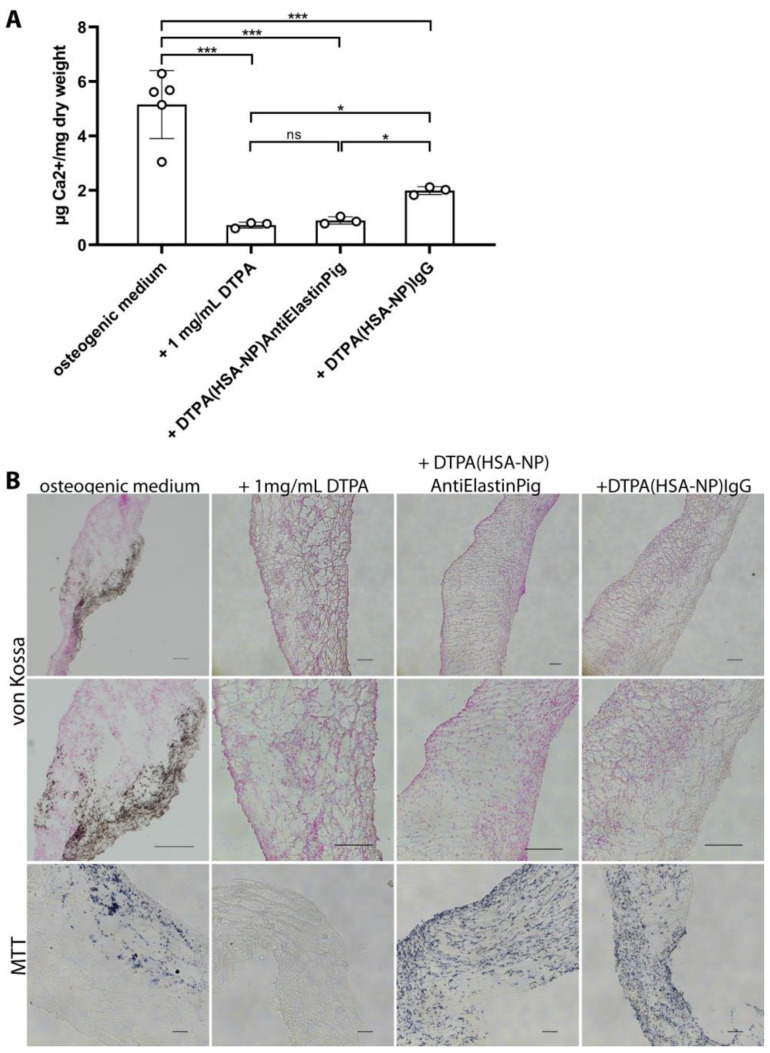
Reversion of induced calcification in porcine aortic valve leaflets using NPs. Leaflets were pre-incubated in osteogenic medium for five days to induce calcification prior to administration of 1 mg/mL DTPA, the equivalent of DTPA bound to DTPA(HSA-NP)AntiElastinPig or DTPA(HSA-NP)IgG as control for two days. (**A**). Quantification of calcium showing reduced calcium content in leaflets incubated with pure DTPA, DTPA(HSA-NP)AntiElastinPig, and DTPA(HSA-NP)IgG compared to control only cultured in osteogenic medium. Mean ± SD; *n* ≥ 3; ((**B**), upper two panels). Representative images of von Kossa staining of aortic valve leaflets visualizing calcified areas (brown) in leaflets incubated in osteogenic medium, while no calcification was found in leaflets treated with pure DTPA, DTPA(HSA-NP)AntiElastinPig, and DTPA(HSA-NP)IgG (middle panel represents magnification of upper panel, scale bars represent 100 µm; *n* ≥ 3 valve leaflets, *n* ≥ 3 sections per leaflet analyzed). ((**B**), lower panel). Approximately half of the cells were viable in leaflets cultured under osteogenic culture conditions. No viability was detectable in leaflets incubated with pure DTPA, whereas leaflets administered with nanoparticle formulations showed viability comparable to leaflets incubated under standard conditions (Figure 1) (scale bars represent 100 µm; *n* ≥ 3 valve leaflets, *n* ≥ 3 sections per leaflet analyzed). * *p* ≤ 0.05; *** *p* ≤ 0.001; ns: not significant.

**Table 1 ijms-24-16471-t001:** Physicochemical characteristics of prepared nanoparticles (mean ± SD; *n* ≥ 3).

Nanoparticle System	Hydrodynamic Diameter [nm]	Polydispersity Index	Zeta Potential [mV]	Drug Load [mol DTPA/mol HSA]
DTPA-(HSA-NP)-AntiElastinPig DTPA-(HSA-NP)-IgG	159.7± 16.5	0.06 ± 0.01	−33.7 ± 1.9	9.0 ± 3.0
(HSA-PF633P-NP)-AntiElastinPig (HSA-PF633P-NP)-IgG	169.4 ± 17.9	0.04 ± 0.03	−41.3 ± 6.0	

## Data Availability

Data are contained within the article.

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
