# Peer review of "Improved Reversion of Calcifications in Porcine Aortic Heart Valves Using Elastin-Targeted Nanoparticles"

_ijms, 2023, doi:10.3390/ijms242216471_

Round 1
Reviewer 1 Report
Comments and Suggestions for Authors
The manuscript describes the application of nanoparticles based on human serum albumin coupled to DTPA a calcium chelator and an elastin antibody (also specific to porcine molecule) on static in vitro aortic valve tissue culture of whole cusps. Aim of the study is to proof the concept of the application of this nanoparticle construct as medical therapy to reverse CAVD.
Methodology and results are insufficient for proof of the hypothesis. The following points lead to the decision to decline the manuscript.
- How was culturing performed? The whole cusps were cultured in a static condition (no shaking?) in a cell culture plate? Was there any evaporation especially important for the calcification induction media leading to precipitation?
- Did the tissues roll in? What impact had these changes? So far as I know the AV tissues have to be spanned to not enrol, that leads to an insufficient medium supply. If there was no enrolment which side was placed to bottom and does this impact the calcification process.
- Was there a viability stain (MTT) quantification? As shown in Figure 1B, parts of the tissue segments are dead. Also in the standard medium after 5 days. Why no control in standard medium at day 10? Counterstain using nuclei dye such as DAPI allows the visualisation of remaining nuclei of death cells.
- No native control (without culturing, directly after slaughtering)
- Van Kossa staining is not in typical black? Kind of brown? Why?
- How was precutting for histological evaluation performed?
- It is recommended not to use Excel for preparation of diagrams but programs such as GraphPadPrism
- 1A: use two Way ANOVA to test all conditions against each other. Or separate the datasets for using a One-Way-ANOVA. Why is for example comparison between day 1 osteogenic medium significantly different from day 7 but not from day 10??? So this calcification is not stable? Is a change from 2 to 5 µg calcium per mg dry weight really a calcification process or just due to penetration of phosphate ions from buffer?
- Figure 2: Was RAMAN analysis performed on three different samples? Why a sample is shown with very low von Kossa staining (not black!). peak at 960 1/cm is very low. Where do phosphates occur in cellular context. What is shown in Y-axis? What is shown in the heat map? Which staining was performed in B? Which sample was investigated? Day 10?
- Why was Raman performed from Parafin embedded sample and not of a cryocut sample with less limitations for this analysis?
- Figure 3: Visualization of particle localisation in Immunofluorescence shows very low positivity in the AntielastinPig sample and a not typical morphology for elastin fibers as they occur in AV tissue (long very thin fibers). This is also the case in Part B of the figure. Why is there no penetration in the standard medium control sample? In the discussion part an antibody is mentioned that binds “core elastin” occurring when elstin is “damaged”. Did You use such an AB? Is a quantification of the staining possible to better estimate the binding in multiple samples?
- Figure 4: A How was statistical testing performed? Use One-way ANOVA to test multiple parallel conditions. Why is there no significant difference between osteogenic medium vs. 1mg/ml DTPA or DTPA….Pig but between osteogenic medium and DTP…IgG? The significant effect semms to be independent from antibody specificity! Isotype also is effective (see values of the control medium in figure 1!!!!). So the concept is not proven!
- Does DTPA impact the pH of the medium? How was it added to the culture media?
- Figure 4 B: no parallel sections provided – unclear viability (morphology of MTT picture in the Osteogenic medium arm). Very weak calcification (bown colour in van Kossa).
- No description or discussion on fabrication of the fabrication of the Nanoparticles Table 1 not mentioned in results or discussion section.
- Elastica van Giesson staining in M&M à not shown; Would be a good staining to visualize the elastin fibres in parallel
- AV of how many pigs were used? Were parallel samples for all investigated conditions used? (Impact of individual)
- Abbreviation in the title
- In the introduction some rare hereditary syndromes are mentioned. Why is this of relevance for this concept?
- Detailed discussion of effects of nanoparticles on other tissues than AV should be provided
- Pathophysiological process of CAVD focusses on impact of lipoprotein. This is only one factor. Processes should be presented more balanced.
Comments on the Quality of English Languagecan be improved
Reviewer 2 Report
Comments and Suggestions for Authors
Feldmann et al. aimed to evaluate the effect of chelator- and anti-elastin antibody bi-conjugated albumin-based nanoparticles on the regression of ex vivo calcification in porcine aortic valves. They found the nanoparticles to reduce aortic valve calcification without significantly affecting cellular viability. The potential of chelators to treat cardiovascular disease has been explored. Still, this work represents a step forward by conjugating an anti-elastin antibody that improves the specificity of the therapy. The study is well-designed, the needed controls are provided, and the results support the conclusion. This paper provides solid evidence to test the nanoparticle formulation in animal models. I have only minor points that the authors can easily addressor points:
0. Abstract
a. Lines 21-22. Confuse phrasing. ‘Anti-elastin targeted’ is redundant. Adding hyphens to “DTPA coupled” and “HSA based NPs” may facilitate interpretation.
b. Line 26. Please remove ‘without significant side effects’. With the simplified ex vivo model used, the absence of off-target effects (for instance, in bone tissue) could not be demonstrated
1. Introduction
a. Lines 47 and 54. For clarity, “Elastin” could be better defined as “Elastin fibres” or “Elastin layers”, provided the mentioned crosslinks and heterogenous glycoproteic contents.
2. Results
a. Line 113. ‘remained their viability’ should be ‘remained viable’
b. Line 114. Define MTT.
c. Lines 126-128 would be better framed after line 114.
d. Figure 4a. Significance levels are not congruent with text descriptions (182-188) or with the definitions given in Methods.
e. Figure 4b. If middle panel images are magnifications of upper panel ones, then not all scale bars represent 100 µm. Please amend.
3. Discussion
a. Lines 213-215. Long confusing sentence. I suggest breaking it down into two sentences to improve clarity.
b. Line 247. ‘leaded’ should be ‘led’
c. Lines 279-281 Confuse wording. I suggest changing it to “We found the anti-elastin antibody-bound NPs (…) to target only already calcified tissue (…)”
d. Line 313. Nanoparticular should be “nanoparticle”.
4. Material and Methods
a. Table 1. I suggest to give PdI in full. The meaning of the “b” letter in superscript is not given.
Comments on the Quality of English LanguageNo significant issues with English. Some sentences would benefit from rewriting, which I marked as minor points.
Reviewer 3 Report
Comments and Suggestions for Authors
In this study, Anja Feldmann and her colleagues have undertaken a comprehensive exploration of elastin-targeted DTPA-HSA nanoparticles for addressing calcifications in porcine aortic heart valves. The study demonstrates the remarkable effectiveness of these nanoparticles in resolving calcifications, offering a promising avenue for targeted nanoparticle-based therapies in the context of aortic valve calcifications. Aortic valve calcification represents a significant impediment to heart valve function, with surgical intervention currently serving as the primary treatment option. This investigation by Anja Feldmann et al. delves into the realm of nanomedicine and targeted drug delivery as prospective therapeutic approaches for reversing calcification. Importantly, the utilization of ex vivo tissue culture, particularly involving porcine aortic valve leaflets, ensures the preservation of physiological arrangements, enhancing the translational potential of these findings.
The utilization of chelating agents such as DTPA holds considerable promise in the realm of cardiovascular diseases. Nevertheless, it's crucial to acknowledge certain limitations in the direct applicability of these findings to human aortic valves, stemming from inherent microstructural disparities.
An exciting prospect for future research lies in transitioning to in vivo experiments. A potential avenue could involve the use of a valvular calcification-induced mouse model, allowing for a more holistic exploration of the therapeutic potential of these nanoparticles within a living organism.
Additionally, I recommend a couple of minor modifications:
1. Ensure that the legends of each figure include information regarding the number of sections or tissues involved.
2. Consider incorporating images of cultured aortic valve leaflets on the culture dish, both control and treated, before and after incubation in osteogenic medium. These images, when stained with Alizarin red, can provide valuable visual insight into the effects of the treatment. Feel free to use this revised comment or make any further adjustments as needed.
Round 2
Reviewer 1 Report
Comments and Suggestions for Authors
The manuscript describes the application of nanoparticles based on human serum albumin coupled to DTPA a calcium chelator and an elastin antibody (also specific to porcine molecule) on static in vitro aortic valve tissue culture of whole cusps. Aim of the study is to proof the concept of the application of this nanoparticle construct as medical therapy to reverse CAVD.
Most of the points from the first review have been implemented and manuscript has been improved.
Still the following points should be further edited:
Figure 4A à all significant differences should be presented (also the significant difference between calcified control vs. Isotype-nanoparticle and DTPA alone. Please also check other figures.
Limitations of static tissue culture and the folding of the material should be implemented in discussion section as well. This can clearly impact the results dependent on which part of the cultured leaflet was used for the analysis of tissue viability and/or calcification! Please also describe in the materials and methods section.
Still the heading should be improved since also the Isotype control results in a significant reduction. “Improved reversion of calcification by elastin-targeted…. As a suggestion
I recommend the LDH viability stain of parallel cryosections of the aortic valve tissue to better discuss the impact of culture viability on calcification process. In addition I recommend the quantification of the “death” area.
Implement change of the pH in culture medium after addition of DTPA in discussion section.
RAMAN signal is very weak and can also occur from cellular context especially in the endothelial layer area with higher cell density as shown in the figure. Did you check a native sample without calcification as well?
Comments on the Quality of English Language-
Author Response
Please find our reply attached.
Regards,
Frank Rutsch

Round 3
Reviewer 1 Report
Comments and Suggestions for Authors
The manuscript describes the application of nanoparticles based on human serum albumin coupled to DTPA a calcium chelator and an elastin antibody (also specific to porcine molecule) on static in vitro aortic valve tissue culture of whole cusps. Aim of the study is to proof the concept of the application of this nanoparticle construct as medical therapy to reverse CAVD.
Most of the points from the first review have been implemented and manuscript has been improved.
Still the following point should be further edited:
Regarding the point from review 2:
I recommend the LDH viability stain of parallel cryosections of the aortic valve tissue to better discuss the impact of culture viability on calcification process. In addition I recommend the quantification of the “death” area.
The Literature implemented so far (39) describes the LDH measurement from media supernatants. This has to be critically discussed as well, since ratio of cells in the tissue to medium volume does perhaps not offer the opportunity to measure low LDH amounts and a death tissue control is missing….
The staining I recommended is the LDH stain of cryosections after culture period; so the enzyme activity of LDH is shown according to tissue distribution of viable cells in the end of the culture period. Viability of tissue can also be revealed by resazurin reduction assay during the culture period. It would be further improve the manuscript to discuss the application of the LDH STAIN of endpoint tissue sections instead of parallel tissue cultures using the MTT assay.
Comments on the Quality of English Language-